# Single-cell atlas of AML reveals age-related gene regulatory networks in t(8;21) AML

Jessica Whittle[1,2], Stefan Meyer[3,4,5], Georges Lacaud[2]*, Syed Murtuza-Baker[1]*, Mudassar Iqbal[1]*

[1]Division of Informatics, Imaging and Data Sciences, Faculty of Biology, Medicine and Health, The University of Manchester, Manchester, United Kingdom; [2]Stem Cell Biology Group, Cancer Research UK Manchester Institute, The University of Manchester, Manchester, United Kingdom; [3]Manchester Cancer Research Centre (MCRC), Division of Cancer Sciences, School of Medical Sciences, Faculty of Biology, Medicine and Health, The University of Manchester, Manchester, United Kingdom; [4]Department of Paediatric and Adolescent Oncology, Royal Manchester Children's Hospital, Manchester, United Kingdom; [5]Department of Adolescent Oncology, The Christie NHS Foundation Trust, Manchester, United Kingdom

**\*For correspondence:**
georges.lacaud@manchester.ac.uk (GL);
syed.murtuzabaker@manchester.ac.uk (SM-B);
mudassar.iqbal@manchester.ac.uk (MI)

**Competing interest:** The authors declare that no competing interests exist.

## eLife Assessment

This manuscript provides a single-cell transcriptomic atlas for AML (222 samples comprising 748,679 cells) integrating data from multiple studies. They use this dataset to investigate t(8;21) AML, and they reconstruct the Gene Regulatory Network and enhancer Gene Regulatory Network, which allowed identification of interesting targets. This aggregation is **important** and can help infer differences in genetic regulatory modules based on the age of disease onset. Their **compelling** effort may help explain age-related variations in prognosis and disease development in subtype-specific manner.

**Abstract** Acute myeloid leukemia (AML) is characterized by cellular and genetic heterogeneity, which correlates with clinical course. Although single-cell RNA sequencing (scRNA-seq) reflects this diversity to some extent, the low sample numbers in individual studies limit the analytic potential when comparing specific patient groups. We performed large-scale integration of published scRNA-seq datasets to create a unique single-cell transcriptomic atlas for AML (AML scAtlas), totaling 748,679 cells, from 159 AML patients and 51 healthy donors from 20 different studies. This is the largest single-cell data resource for human AML to our knowledge, publicly available at https://cellxgene.cziscience.com/collections/071b706a-7ea7-47a4-bddf-6457725839fc. This AML scAtlas allowed investigations into 20 patients with t(8;21) AML, where we explored the clinical importance of age, given the in-utero origin of pediatric disease. We uncovered age-associated gene regulatory network (GRN) signatures, which we validated using bulk RNA sequencing data to delineate distinct groups with divergent biological characteristics. Furthermore, using an additional multiomic dataset (scRNA-seq and scATAC-seq), we validated our initial findings and created a de-noised enhancer-driven GRN reflecting the previously defined age-related signatures. Applying integrated data analysis of the AML scAtlas, we reveal age-dependent gene regulation in t(8;21) AML, potentially reflecting immature/fetal HSC origin in prenatal origin disease vs postnatal origin. Our analysis revealed that BCLAF1, which is particularly enriched in pediatric AML with t(8;21) of inferred in-utero origin, is a promising prognostic indicator. The AML scAtlas

provides a powerful resource to investigate molecular mechanisms underlying different AML subtypes.

## Introduction

Acute myeloid leukemia (AML) is an aggressive blood cancer driven by non-random genomic rearrangements in hematopoietic stem/progenitor cells (HSPCs). Recurrent AML-associated genomic aberrations, which often involve transcriptional or epigenetic regulators, give rise to distinct patterns of gene expression strongly associated with clinical course and chemotherapy response (*Tenen, 2003*; *Bolouri et al., 2018*). Single-cell RNA sequencing (scRNA-seq) studies have demonstrated that HSPCs acquire lineage priming at an early stage when still phenotypically immature and disperse down an erythromyeloid or lymphomyeloid differentiation trajectory (*Velten et al., 2017*). In the context of AML, diverse clonal hierarchies include the co-existence of normal hematopoietic clones. Leukemic clones can partially recapitulate myeloid differentiation and have been shown to display functional differences even when defined by the same genotype (*Velten et al., 2021*; *van Galen et al., 2019*; *Beneyto-Calabuig et al., 2023*; *Zeng et al., 2022*). Indeed, analysis of AML using scRNA-seq has revealed key clonal hierarchies, defining subtype-associated cell types, and dynamic changes following therapy, and have been critical in characterizing leukemic stem cells (LSCs), which propagate the disease and drive relapse (*van Galen et al., 2019*; *Beneyto-Calabuig et al., 2023*; *Zeng et al., 2022*; *Stetson et al., 2021*).

Most AML scRNA-seq studies are limited by small sample numbers and include a mixture of different AML subtypes which may not be directly comparable to one another. Therefore, it is difficult to make biological conclusions with sufficient robustness to be clinically translatable in these individual datasets. To overcome this, we performed large-scale integration of public scRNA-seq datasets to create a single-cell transcriptomic atlas for AML (AML scAtlas). Due to the range of data sources spanning time, locations, and experimental designs, complex batch effects often arise between scRNA-seq datasets, which requires a tailored data integration approach (*Luecken et al., 2022*; *Heumos et al., 2023*). Thus, we benchmarked some widely used batch correction tools (*Korsunsky et al., 2019*; *Lopez et al., 2018*; *Xu et al., 2021*) for our specific data use case.

Given the broad representation of age groups in AML scAtlas, we sought to investigate a developmental aspect of AML biology. Pediatric AML has substantially better clinical outcomes compared to adult AML (*Balgobind et al., 2011*; *Wiggers et al., 2019*; *Chaudhury et al., 2018*). The molecular landscape of AML differs between children and adults (*Bolouri et al., 2018*; *Balgobind et al., 2011*; *Wiggers et al., 2019*; *Chaudhury et al., 2018*) this may, in part, reflect differences in the developmental origins of the disease. Chromosomal changes in pediatric leukemia are acquired in-utero, as evidenced by leukemia-specific genomic aberrations detected in the Guthrie spots of children who later developed leukemia, sometimes several years after birth (*Wiemels et al., 2002*). Adult leukemia, in contrast, is thought to develop later in life through acquisition of pre-leukaemic changes and clonal evolution of adult HSPCs (*Welch et al., 2012*; *Jaiswal et al., 2014*). The impact of these developmental stages on leukemia biology remains incompletely understood, and no current methods exist to quantify and characterize differences in the origin of the disease. However, as childhood AML with presumed in-utero origin has a better outcome, for teenagers and young adults, determination of the pre- or postnatal origin might be important for better treatment stratification and prognostication.

AML with t(8;21) (AML-ETO/RUNX1-RUNX1T1) is one of the most frequent AML subtypes in young people, although it affects all ages (*Bolouri et al., 2018*). The prenatal origins of the t(8;21) rearrangement has been confirmed even in older children presenting with AML (*Wiemels et al., 2002*). The prognosis of AML with t(8;21) is better in children than in teenagers and even more so than in young adults (*National Cancer Registration and Analysis Service, Northern Ireland Cancer Registry, Scottish Cancer Registry, Unit WCIaS, 2021*). This outcome difference is not fully explainable by co-morbidities and may instead be related to the developmental origins of the disease. In the intermediate teenage group, t(8;21) AML may comprise both late childhood and early adult disease entities, a distinction that could have prognostic implications and could help to explain disease biology and clinical course.

We leveraged our AML scAtlas resource to characterize age and developmental stage-specific signatures in t(8;21) AML by applying single-cell gene regulatory network (GRN) inference (*Aibar*

*et al., 2017*; *Van de Sande et al., 2020*), as a means of revealing cell state heterogeneity across age groups. We then validated and refined our findings in a larger cohort using bulk RNA sequencing (RNA-seq) data from the TARGET (*Bolouri et al., 2018*) and BeatAML (*Burd et al., 2020*) studies, defining age-associated GRN signatures and key regulators of t(8;21) AML, that may reflect the developmental origins of the leukemia.

Profiling both gene expression and chromatin accessibility together can decipher the enhancer-driven GRN (eGRN) and enriched transcriptional regulators. Significant heterogeneity across different patients and time points (*Lambo et al., 2023*) was recently described by analyzing combined scRNA-seq and single-cell Assay for Transposase Accessible Chromatin sequencing (scATAC-seq). We used the t(8;21) AML data from this study to validate our initial findings, by applying cutting edge GRN inference methodology (*Bravo González-Blas et al., 2023*). This encompasses both modalities to provide a denoised eGRN which we could correlate with our age-associated signatures.

## Results

### Large scale data integration to construct a single-cell transcriptomic atlas of AML (AML scAtlas)

To create the AML scAtlas, we integrated published scRNA-seq data of primary AML bone marrow samples, from 16 suitable high-quality studies (see Materials and methods), comprising 159 AML samples (*Figure 1A*; *Source data 1*). Where on-treatment time points were available, we selected only diagnostic samples to establish a reference atlas of primary AML at diagnosis. If studies had healthy donor bone marrow samples, these were included, alongside data from healthy bone marrow samples from four additional scRNA-seq studies (*Source data 1*) to enable comparisons between malignant and healthy bone marrow populations. After cell filtering and quality control, the AML scAtlas contains data from 748,679 high quality cells derived from a total of 20 different scRNA-seq studies (*Velten et al., 2021*; *van Galen et al., 2019*; *Beneyto-Calabuig et al., 2023*; *Stetson et al., 2021*; *Zheng et al., 2017*; *Petti et al., 2019*; *Jiang et al., 2020*; *Johnston et al., 2020*; *Pei et al., 2020*; *Li et al., 2023*; *Lasry et al., 2023*; *Fiskus et al., 2023*; *Naldini et al., 2023*; *Mumme et al., 2023*; *Zhang et al., 2023*; *Li et al., 2022*; *Oetjen et al., 2018*; *Setty et al., 2019*; *Caron et al., 2020*; *Figure 1—figure supplement 1A*). Each sample was assigned to an AML clinical subtype, based on the recent European Leukemia Net (ELN) clinical guidelines (*Döhner et al., 2022*), and classified into the corresponding prognostic risk group. This resource captures a broad range of molecular subtypes of AML and spans different age groups, including both pediatric and adult AML cases (*Figure 1B–C*). Overall, this is the largest dataset to date for exploring AML biology at single-cell resolution.

In the initial analysis of the combined dataset, batch effects were noted with study-specific clustering, which was quantified using several benchmarking metrics (*Figure 1—source data 2*; *Figure 1—figure supplement 1A and B*). Even within samples of the same study, sample-wise clustering was noted (*Figure 1—figure supplement 1C*). To address this, we benchmarked several widely used batch correction methods (*Figure 1—source data 2*; *Figure 1—figure supplement 2A–C*), identifying scVI as the best method for this dataset (*Figure 1—source data 2*). We, therefore, employed scVI to correct for batch effects, before clustering and cell type annotation in the AML scAtlas, by using the consensus of multiple annotation tool results (*Figure 1—source data 2*), verified using cluster-wise marker gene expression (*Figure 1D–E*).

Cell type proportions analyses across the clinically relevant subtypes in the dataset show that the AML subtypes were significantly biased towards myeloid cell types (CMP, MEP, GMP, ProMono, CD14+ Mono, CD16+ Mono, cDC, Erythroid) with each subtype exhibiting a clear predominant cell type consistent with AML clonal expansion (*Figure 2A*; *Figure 2—figure supplement 1A–B*). In contrast, healthy donor samples had more balanced lineage proportions, with lymphoid cells (T, B, NK, ProB, pDC, Plasma) well represented (*Figure 2A*; *Figure 2—figure supplement 1A–B*). Given the established critical role of HSPCs and LSCs in propagating AML, and their importance as therapeutic targets (*Montefiori et al., 2021*), we focused on HSPC clusters for further analysis (*Figure 2B*). To identify LSCs, we applied a curated reference profile of leukaemic stem and progenitor cells (LSPCs) (*Zeng et al., 2022*; *Figure 2B–C*) and correlated this with calculated LSC6 (*Elsayed et al., 2020*) and LSC17 (*Ng et al., 2016*) scores for each cell (*Figure 2D*). We then compared the proportions of HSPC/LSPCs across different AML subtypes and risk groups, as defined by the ELN clinical guidelines (*Döhner*

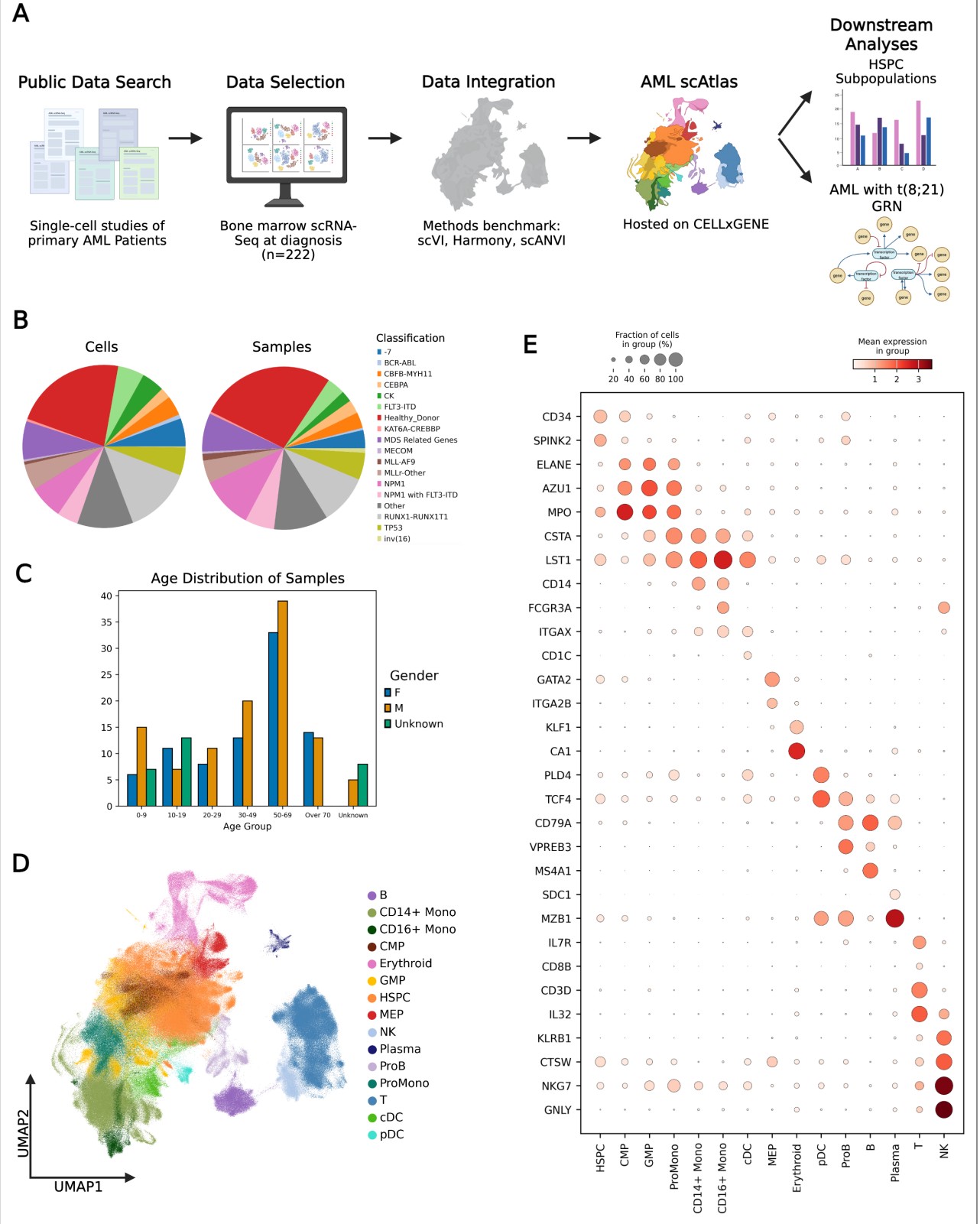

**Figure 1.** Large scale data integration creates a single-cell atlas of acute myeloid leukemia (AML). (**A**) Overview of the analysis steps in creating AML scAtlas. (**B**) Proportion of cells (left panel) and samples (right panel) belonging to each AML subtype as defined by the European Leukemia Net (ELN) clinical guideline. (**C**) Age group and gender distribution of AML single-cell Atlas (scAtlas) cohort samples. (**D**) scVI harmonized UMAP colored by

*Figure 1 continued on next page*

*Figure 1 continued*

annotated cell types. (**E**) The expression of key hematopoietic marker genes across annotated cell types shown on a dotplot. Color scale shows mean gene expression, dot size represents the fraction of cells expressing the given gene.

The online version of this article includes the following source data and figure supplement(s) for figure 1:

**Source data 1.** Batch correction benchmarking metrics.

**Source data 2.** Automated cell type annotation results.

**Figure supplement 1.** Initial analysis establishes presence of batch effects.

**Figure supplement 2.** Benchmarking batch correction methods.

*et al., 2022*; *Figure 2E*). Higher-risk subtypes displayed a higher proportion of LSCs compared to favorable risk disease (*Figure 2E–F*).

## Application of AML scAtlas to identifying age-associated gene regulatory networks in t(8;21) AML

The AML scAtlas enables robust comparison of adult and pediatric AML. We hypothesized that in adolescents and young adults with t(8;21) AML, the potential for either in-utero or postnatal HSPC origin disease might affect disease biology and prognosis. Thus, we sought to explore biological differences between pediatric and adult cases of t(8;21) AML, aiming to explain and potentially improve prognostication in adolescents and young adults. We selected samples with t(8;21) AML from the AML scAtlas, resulting in 105,663 cells from 13 adult cases (aged 20–67), seven adolescent cases (aged 12–17), and three pediatric cases (aged 6–8) (*Figure 3A–C*). Where gender information was not available, this was inferred from ChrY/XIST gene expression (*Figure 3—figure supplement 1A*). Several adult samples underwent CD34 selection in original studies, excluding more differentiated cell types (mature lymphoid populations, monocytes, granulocytes) in these samples. Thus, these cell types were excluded from comparative analysis, focusing only on HSPCs and myeloid progenitors (CMP, GMP, MEP), which were well represented in all studies (*Figure 3C*).

We reconstructed the GRN for the t(8;21) subset using the pySCENIC (*Van de Sande et al., 2020*) pipeline, which is a Python-based efficient implementation of original SCENIC method (*Aibar et al., 2017*). It is a state-of-the-art method for network inference from scRNA data, popular in the community (*Hamed et al., 2022*; *Barnett et al., 2024*; *Zhang et al., 2024*) and has shown strong performance in a recent benchmarking study (*Nguyen et al., 2021*). SCENIC's three major steps are: First, it identifies groups of co-expressed genes as potential targets of a transcription factor (TF). Second, it refines these groups of genes based on the enrichment of the corresponding TF binding motif, forming 'regulons.' Third, it uses the AUCell method (embedded within SCENIC) to quantify the activity of each regulon in every cell. AUCell calculates the Area Under the Curve (AUC) for the regulon's genes set in a ranking of all genes by expression for each cell. The top 20 regulons for each age groups were selected based on the regulon specificity score (RSS) (*Figure 3—figure supplement 1B*). Unsupervised clustering on the Zscore normalized regulon activity score matrix revealed clear differences in the GRN across different age groups (*Figure 3—figure supplement 1C*). We hypothesize that the differences in the GRN might reflect differences in the pre- or postnatal developmental origins of the disease. Additional testing of GRN inference from individual studies shows that the high number of cells refines the overall GRN (see Methods; *Figure 3—figure supplement 1D–E*).

To define gene regulatory programs (co-occurring gene modules, defined by a transcription factor and its targets) which are specific to different age groups (termed 'regulon signature'), we used the clustered dendrogram to select the regulon clusters most associated with the pediatric (below 10 years-old) and adult samples (over 18 years-old) (*Figure 3D*; *Figure 3—figure supplement 1C*). The pediatric regulon signature, proposed to represent in-utero origin t(8;21) AML (henceforth termed 'inferred-prenatal'), includes 16 regulons defined by a distinct group of hematopoietic transcription factors (TFs) (TRIM28, CTCF, RAD21, SOX4, TAL1, MYB, FOXN3, JUND, BCLAF1, ZBTB7A, IKZF1, MAZ, REST, YY1, CUX1, KDM5A), many of which have clearly defined roles in HSPCs and AML (*Lu et al., 2018*; *Fisher et al., 2017*; *Kumar et al., 2023*). The adult regulon signature, presumed representative of the postnatally acquired t(8;21) AML (henceforth termed 'inferred-postnatal'), combines three discrete clusters of regulons (YBX1, ENO1, and HDAC2; GATA1, POLE3, TFDP1, MYBL2, E2F4, and KLF1; IRF1, STAT1, IRF7, MAFF, ATF4, TAGLN2, SPI1, and KLF2), defined by TFs previously

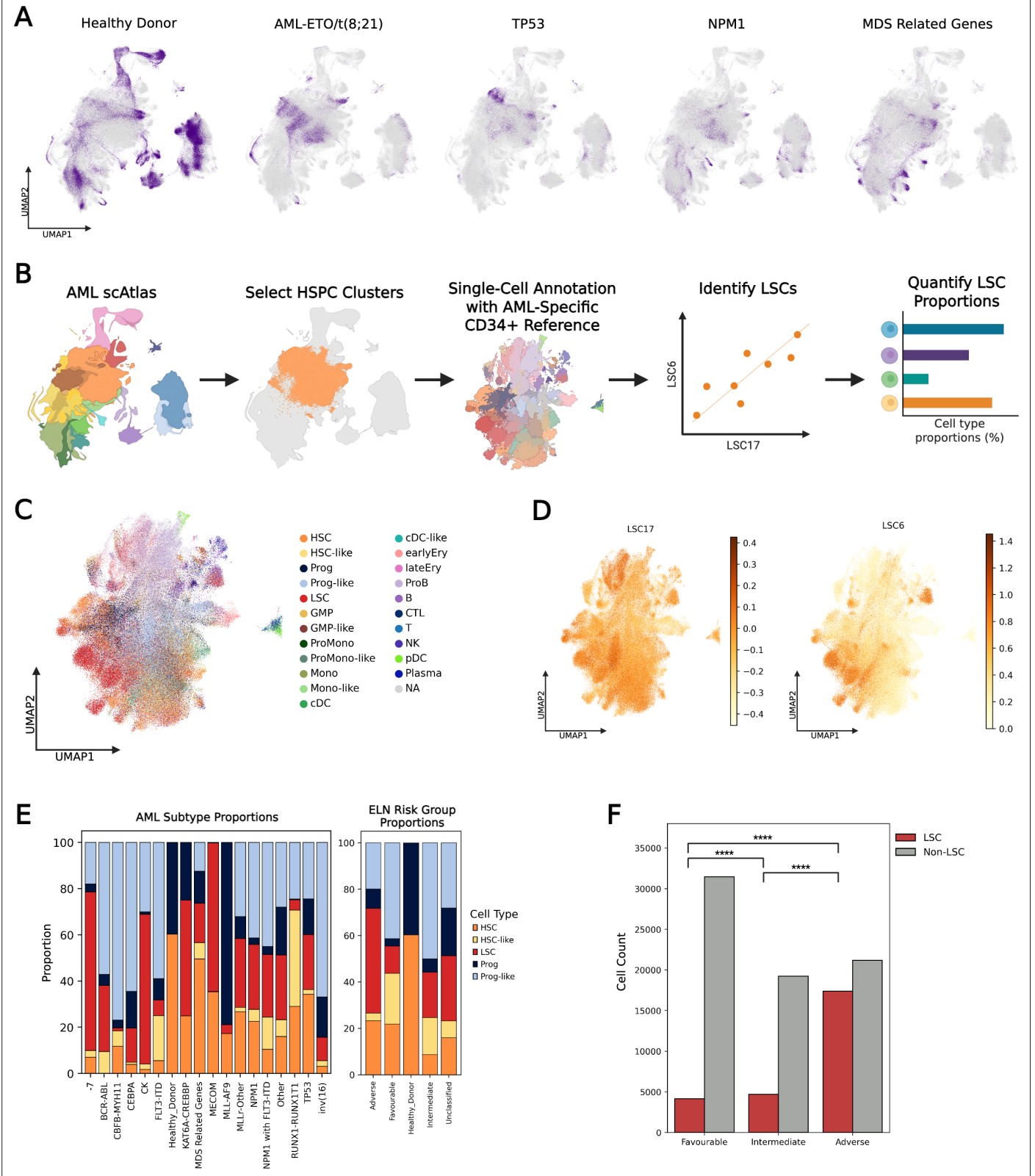

**Figure 2.** Characterizing cell type distributions in acute myeloid leukemia (AML) subtypes. (**A**) UMAP highlighting the distribution of cells from different AML subtypes in AML single-cell Atlas (scAtlas). (**B**) Schematic showing the workflow used to identify leukemic stem cells (LSCs) from the AML scAtlas hematopoietic stem and progenitor cell (HSPC) clusters. (**C**) Using the AML scAtlas HSPC clusters only, UMAP was regenerated and annotated with an AML-specific reference of leukemia stem and progenitor cells (LSPCs). (**D**) UMAPs showing the leukemic stem cell scores of each cell, for the LSC17

*Figure 2 continued on next page*

*Figure 2 continued*

(left) and LSC6 (right). (**E**) Proportions of HSPC/LSPC populations in different AML subtypes (left) and AML risk groups (right), as defined by European Leukemia Net (ELN) clinical guidelines. (**F**) Comparison of LSC abundance in favourable and adverse ELN risk groups. Chi-Square test statistic: 8658.98, degrees of freedom: 1, p-value: 0.0.

The online version of this article includes the following figure supplement(s) for figure 2:

**Figure supplement 1.** Cell type proportions vary by acute myeloid leukemia (AML) subtype.

implicated in various hematopoietic, leukemic and inflammatory processes (*Ning et al., 2011*; *Fischer et al., 2019*; *Fang et al., 2018*). Importantly, both signatures contain key components of the AP-1 complex, which is heavily implicated in the biology of t(8;21) AML (*Ptasinska et al., 2019*; *Martinez-Soria et al., 2018*) and undergoes dynamic changes during aging (*Patrick et al., 2024*). Samples of 6 individuals aged 12–17 clustered with the pediatric samples and showed enrichment for the inferred-prenatal signature (*Figure 3D*), suggesting that older adolescents (up to age 17 in our cohort) more closely resemble pediatric AML with t(8;21) and remain biologically distinct from adult-onset disease. This implies that the inferred in-utero origin of t(8;21) AML can also be present in AML diagnosed in older children.

## Validation of age-associated regulons in bulk-RNA-seq cohorts of t(8;21) AML

We next sought to externally validate our age-associated regulon signatures in a larger cohort of patients. Bulk RNA-seq samples were obtained from the TARGET (*Bolouri et al., 2018*) and BeatAML (*Burd et al., 2020*) cohorts, selecting bone marrow samples taken at diagnosis in line with AML scAtlas data (n=83; *Source data 2*). We applied the AUCell algorithm from pySCENIC (*Van de Sande et al., 2020*) to calculate the activity of our pediatric inferred-prenatal and adult inferred-postnatal regulons in each sample. Unsupervised clustering of the bulk RNA-seq AUCell results revealed discrete clusters of samples that were highly enriched for our inferred-prenatal and inferred-postnatal origin-associated regulons (*Figure 4A*).

Given the limitations of most scRNA-seq platforms in detecting lowly expressed genes, notably TFs, we leveraged bulk RNA-seq samples to refine our identified gene regulatory networks by detecting differentially expressed regulon-associated TFs. We used our inferred-prenatal and inferred-postnatal signature clusters and performed differential gene expression analysis between these samples, using two widely used tools (DESeq2 *Robinson et al., 2010* and edgeR *Love et al., 2014*) to ensure robustness of the results (*Figure 4B*; *Figure 4—figure supplement 1A*). We then compared differentially expressed regulon-associated TFs between the two groups and intersected this with the differential genes detected by each method. Although changes in TF expression are subtle (*Figure 4B*; *Figure 4—figure supplement 1A*), we identify significantly differentially expressed TFs which reflect the observed differences in regulon activity and indicate the most critical regulons in our age-related GRN signatures (*Figure 4A–B*; *Figure 4—figure supplement 1B*). This further delineated the inferred-prenatal signature to five key TFs (KDM5A, REST, BCLAF1, YY1, and RAD21), and the inferred-postnatal signature to eight TFs (ENO1, TFDP1, MYBL2, TAGLN2, KLF2, IRF7, SPI1, and YBX1).

We next performed gene set enrichment analysis (GSEA) on significantly differentially expressed genes as determined by edgeR (*Love et al., 2014*), to investigate pathways enriched in the inferred-prenatal samples compared to the inferred-postnatal ones (*Figure 4C*). Notably, inferred-prenatal samples showed increased expression of stemness-associated genes, and SMARCA2 target genes, a key player in HSC gene expression regulation and chromatin remodeling (*Holmfeldt et al., 2016*). SMARCA2 is also known to be upregulated during the fetal-to-adult HSC transition (*Chen et al., 2019*), implying that the observed SMARCA2 enrichment may indeed reflect the inferred fetal HSC cell-of-origin. Genes impacted by YY1 depletion were also downregulated compared to the samples of inferred-postnatal leukemia origin, which supports the identification of YY1 as an inferred-prenatal regulon (*Figure 4C*). To explore therapeutic implications, we performed GSEA using drug response signatures from published studies (*Unnikrishnan et al., 2017*; *Williams et al., 2020*; *Xu et al., 2019*; *Zhang et al., 2020*; *Figure 4D*). This analysis revealed that inferred-prenatal origin t(8;21)

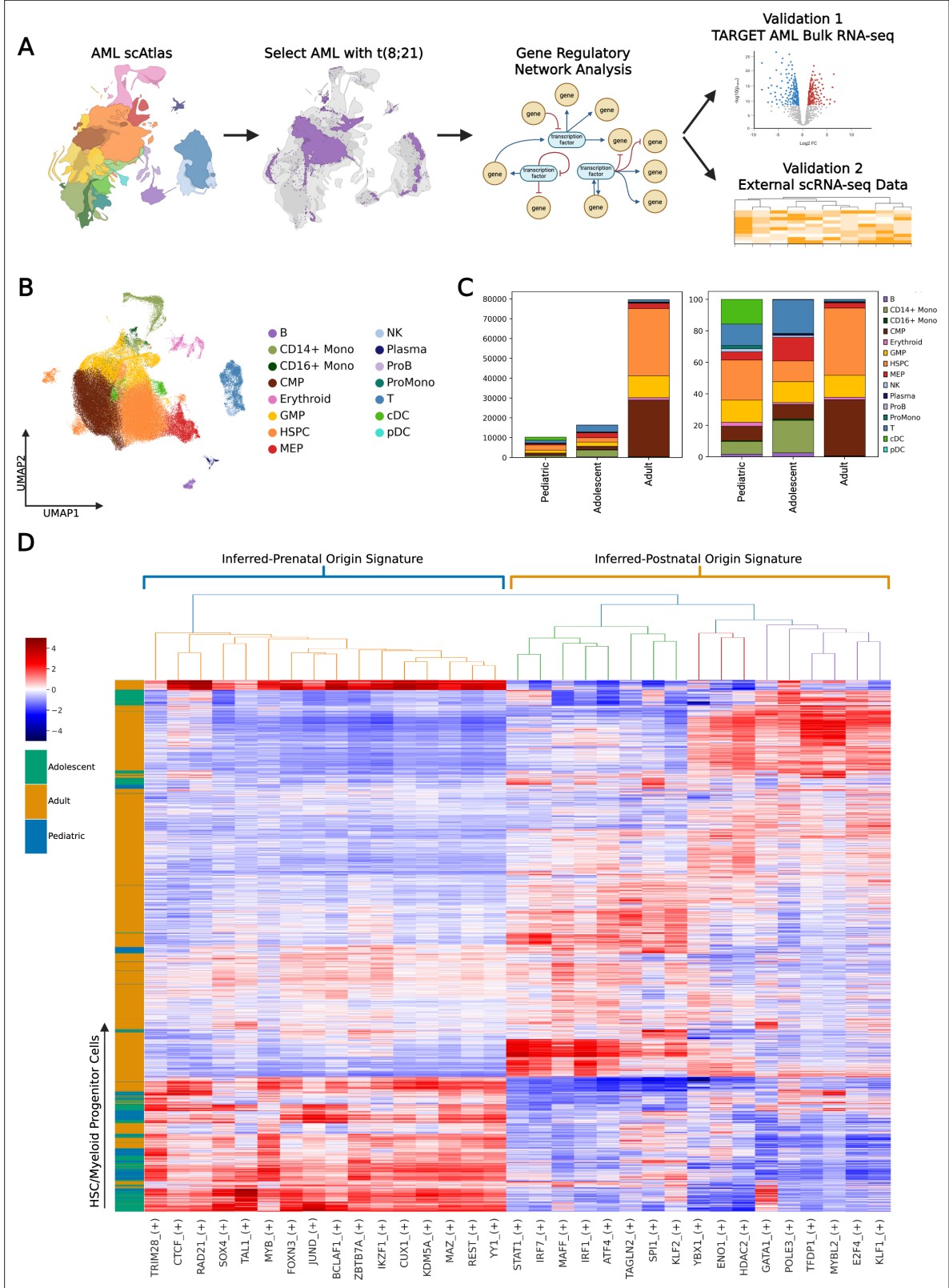

**Figure 3.** Acute myeloid leukemia (AML) single-cell Atlas (scAtlas) reveals age-associated heterogeneity in t(8;21) AML. (**A**) Depiction of the workflow to generate and validate the t(8;21) AML gene regulatory network (GRN) from AML scAtlas. (**B**) Using the AML scAtlas t(8;21) sample cells, UMAP was re-computed and shows the different cell types. (**C**) Bar plots of the absolute cell type numbers (left panel) and the cell type proportions (right panel) stratified by age group. The CD34 enrichment performed on several adult samples is reflected. (**D**) Using HSPCs and CMPs only, the pySCENIC gene

*Figure 3 continued on next page*

*Figure 3 continued*

regulatory network (GRN) and regulon AUC scores were calculated. Z-score normalized scores underwent hierarchical clustering to create a clustered heatmap and identify age-associated regulons. Regulons were prioritized using their regulon specificity scores (RSS).

The online version of this article includes the following figure supplement(s) for figure 3:

**Figure supplement 1.** Acute myeloid leukemia (AML) with t(8;21) pySCENIC analysis.

AML is enriched for genes associated with increased chemosensitivity to cytarabine, venetoclax, and daunorubicin.

We hypothesized that the increase in stemness-associated genes in the leukemia with the inferred-prenatal origin could be reflective of potential differences in the leukemic cell-of-origin and its impact on myeloid differentiation. We therefore performed cell type deconvolution using AutoGeneS (*Aliee and Theis, 2021*), with a curated LSPC reference profile (*Zeng et al., 2022*), to compare the cellular heterogeneity between prenatal and postnatal origin bulk RNA-seq samples (*Figure 4E*). This revealed a higher proportion of HSPC cell types (HSC, Prog), with a reduction in some differentiated myeloid cell types (ProMono-like, cDC-like) in the samples of inferred-prenatal origin (*Figure 4E*). To corroborate this finding, we examined cell type proportions in the original t(8;21) subset of AML scAtlas, confirming that cells with the inferred-prenatal signature comprise more HSCs than inferred-postnatal signature cells (*Figure 4—figure supplement 1D–E*). However, comparison of cell type proportions in this dataset is confounded by differences in sample processing as some studies performed CD34 selection, hence, there is more cell type diversity observed in the pediatric samples (*Figure 4—figure supplement 1D–E*).

## Multiomics single-cell data reveals a denoised GRN and identifies candidate perturbations in prenatal origin t(8;21) AML

We next used the scRNA-seq and scATAC-seq data from a recent cohort of pediatric t(8;21) AML patients (*Lambo et al., 2023*) at multiple clinical time points to uncover the enhancer-driven GRN (eGRN) in inferred-prenatal and inferred-postnatal origin t(8;21) AML (*Source data 2*). Initially, we identified two representative samples of our inferred-prenatal and inferred-postnatal signatures by using pySCENIC AUCell (*Van de Sande et al., 2020*) to measure the activity of our previously defined regulons. Unsupervised clustering of the AUC scores was used to infer whether each sample matched the regulon signatures, identifying one inferred-prenatal sample and one inferred-postnatal sample for downstream analysis (*Figure 5—figure supplement 1A*).

We then applied SCENIC+ (*Bravo González-Blas et al., 2023*), which integrates scRNA-seq and scATAC-seq to identify candidate enhancer regions and TF-binding motifs, linking TFs to target genes and identified enhancers. This creates enhancer-driven regulons (eRegulons), forming an eGRN. We applied SCENIC+ (*Bravo González-Blas et al., 2023*) to the leukemia samples with the inferred-prenatal and inferred-postnatal origin at diagnosis and relapse, keeping only regulons that showed a correlation between both modalities to retain only the most robust regulons (*Figure 5—figure supplement 1B*). This revealed several eRegulons across both patients (*Figure 5—figure supplement 1D*), many of which were patient-specific, particularly when comparing HSPC populations (*Figure 5A*). The inferred-prenatal sample displayed a specific HSC eRegulon profile. In contrast, the inferred-postnatal sample more closely resembled the corresponding Granulocyte-Monocyte Progenitor (GMP) (*Figure 5A*). Interestingly, at relapse the inferred-prenatal origin patient undergoes a chemotherapy-driven lineage switch to a lymphoid phenotype, which may suggest that the leukemia originated from a less committed progenitor (*Figure 5A*).

To identify clusters of closely related eRegulons, we computed the correlations between eRegulons enrichment. We identified two main clusters of eRegulons which correspond to different inferred-signature samples (*Figure 5B*; *Figure 5—figure supplement 1C*). For each eRegulon cluster, we used the associated target genes as input for gene ontology over representation analysis (ORA), to assess functional differences in the eGRN. This revealed fundamental differences in the underlying biological processes (*Figure 5C*; *Figure 5—figure supplement 2A*). The AML sample with inferred-prenatal origin was enriched for many processes associated with development. In contrast, inferred-postnatal samples appeared more metabolism focused (*Figure 5C*; *Figure 5—figure supplement*

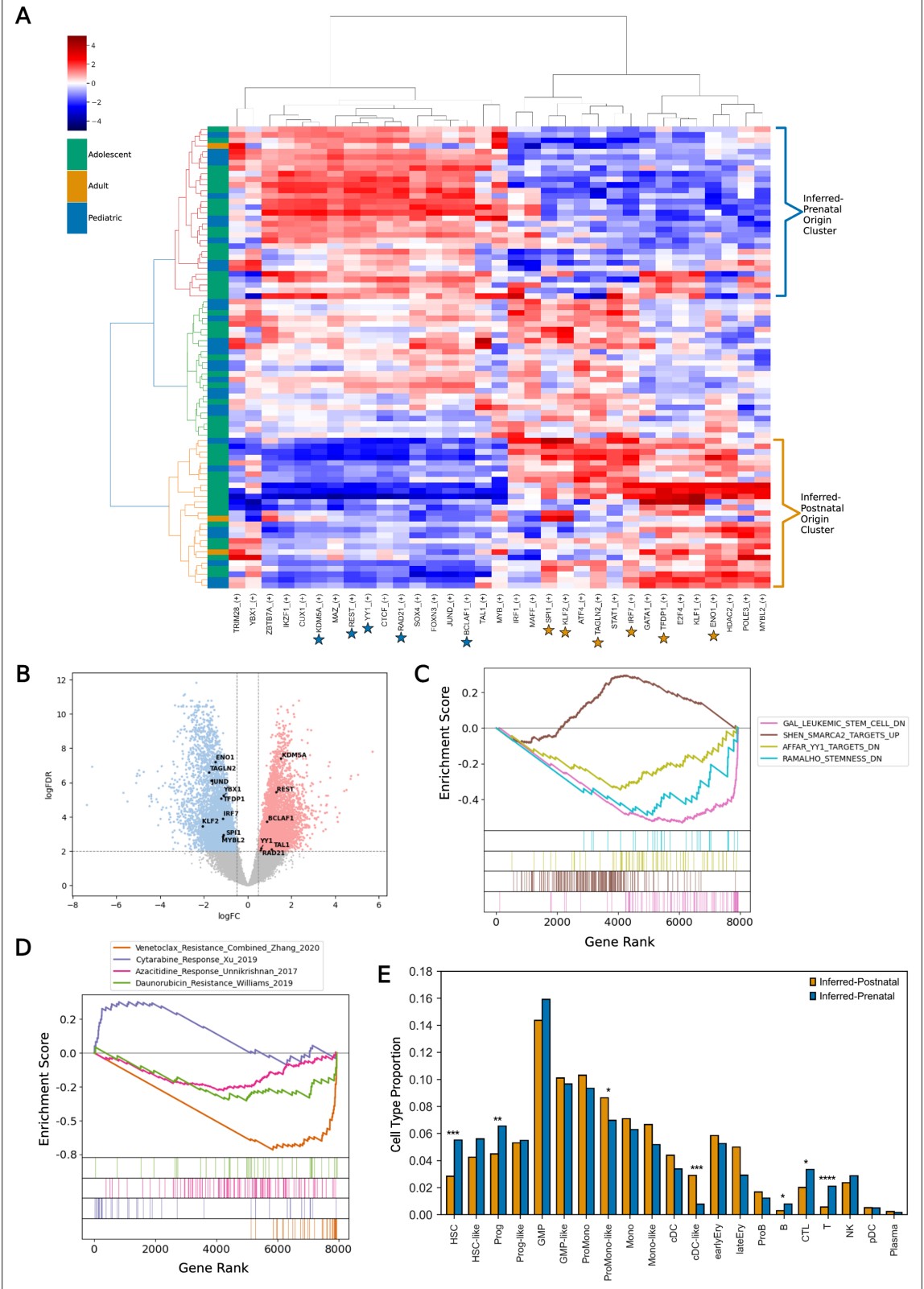

**Figure 4.** Validation of age-associated regulons in large bulk RNA-seq cohorts. (**A**) Using previously defined age-associated regulons, pySCENIC AUC scores (Z-score normalized) were clustered to identify bulk RNA-seq samples (n=83) most enriched for inferred-prenatal and inferred-postnatal origin signatures. (**B**) Volcano plot of differentially expressed genes when comparing the inferred-prenatal origin (n=31) and inferred-postnatal origin (n=27) samples. Adjusted p-value threshold 0.01; log2 fold change threshold 0.5. Regulon signature associated transcription factors (TFs) are

*Figure 4 continued on next page*

*Figure 4 continued*

indicated. (**C**) Enrichment plot of significant gene sets enriched in the inferred-prenatal origin samples. GSEA was performed on the DEGs using MSigDB databases. FDR q-value threshold <0.05. (**D**) Enrichment plot of drug sensitivity gene sets enriched in the inferred-prenatal samples. GSEA was performed on the DEGs, using drug response signatures from published studies of four widely used acute myeloid leukemia (AML) drugs. FDR q-value threshold <0.05. (**E**) The predicted cell type proportions estimated using AutoGeneS deconvolution, of the inferred-prenatal (n=31)and inferred-postnatal origin samples (n=27) were compared using t-tests. Significant *p*-values <0.05 (*), <0.01 (**), <0.001 (***), and <0.0001 (****) are indicated.

The online version of this article includes the following figure supplement(s) for figure 4:

**Figure supplement 1.** Validation of pySCENIC regulons.

*2A*). This further supports the association of these eRegulons with presumed prenatal origin t(8;21) AML, compared to postnatal origin disease.

Previous analysis using the TARGET (*Bolouri et al., 2018*) and BeatAML (*Burd et al., 2020*) datasets indicated that inferred-prenatal and inferred-postnatal origin t(8;21) AML may harbor different levels of chemosensitivity based on published drug response signatures (*Figure 4D*). Therefore, we performed in silico perturbations of eRegulon-associated TFs. PCA of the diagnosis and relapse samples recapitulated the expected differentiation trajectories along PC2, while separating diagnosis from relapse along PC1 (*Figure 5D*). Using the SCENIC+ (*Bravo González-Blas et al., 2023*) perturbation simulation workflow, we identified TFs estimated to induce differentiation, as defined by a negative shift in PC2 (*Figure 5—figure supplement 2B*). We prioritized TFs predicted to impact the HSC compartment and identified 18 TFs with predicted significant effects on HSC differentiation (*Figure 5—figure supplement 2C*). Several of these are components of the AP-1 complex (JUN, ATF4, FOSL2), which are established downstream targets of the t(8;21) fusion protein and are known to propagate t(8;21) AML (*Ptasinska et al., 2019*; *Eferl and Wagner, 2003*; *Figure 5E*; *Figure 5—figure supplement 2C*).

Using AP-1 complex members as a comparative baseline, we identified EP300 as one of the most impactful hits. EP300 has recently been shown to drive t(8;21) AML self-renewal through acetylation dependent mechanism (*Wang et al., 2011*). This suggests that presumed prenatal origin pediatric t(8;21) AML may be particularly sensitive to EP300 inhibition. One of the most striking predictions, for both diagnostic and relapse HSC populations, was BCLAF1 (*Figure 5E*; *Figure 5—figure supplement 2C*). BCLAF1 is a regulator of normal HSPCs (*Crowley et al., 2022a*), and its expression level declines during hematopoietic differentiation. While recent studies have identified a role for BCLAF1 in AML (*Dell'Aversana et al., 2017*), this has not been explored in detail in the context of pediatric AML or t(8;21) AML and may present a therapeutic opportunity.

We also performed SCENIC+ (*Bravo González-Blas et al., 2023*) perturbation modelling on the postnatal origin sample (AML12). In this case, the PCA was less straightforward to interpret, as branching differentiation trajectories towards a lymphoid or myeloid fate appear along the PC2 axis, while PC1 distinguishes diagnosis and relapse samples (*Figure 5—figure supplement 2D*). Therefore, we prioritized TFs based on a predicted effect similar to AP-1 complex components, as it is known that this complex is a critical regulator in t(8;21) AML. We identified several TFs from our original postnatal origin signature were predicted to have an effect (*Figure 5—figure supplement 2D–F*), supporting the relevance of the GRNs identified in our previous analyses.

To further investigate EP300 and BCLAF1, we queried the DepMap (*Tsherniak et al., 2017*) database to assess the dependency of t(8;21) AML cell lines to these genes (*Figure 5—figure supplement 2G*). We found that the two widely used cell lines of t(8;21) AML, KASUMI-1 (7-year-old donor) (*Asou et al., 1991*) and SKNO-1 (22-year-old donor) (*Matozaki et al., 1995*), were among the most sensitive to these perturbations based on their DepMap effect scores (*Figure 5—figure supplement 2G*). Several other cell lines sensitive to BCLAF1 were derived from pediatric cancers, most notably neuroblastomas, which also arise in-utero (*Körber et al., 2023*; *Figure 5—figure supplement 2H*). Together, these findings suggest that EP300 inhibition may be particularly effective in t(8;21) AML, and that BCLAF1 may present a new therapeutic target for t(8;21) AML, particularly in pediatric cases with inferred prenatal origin of the driver translocation.

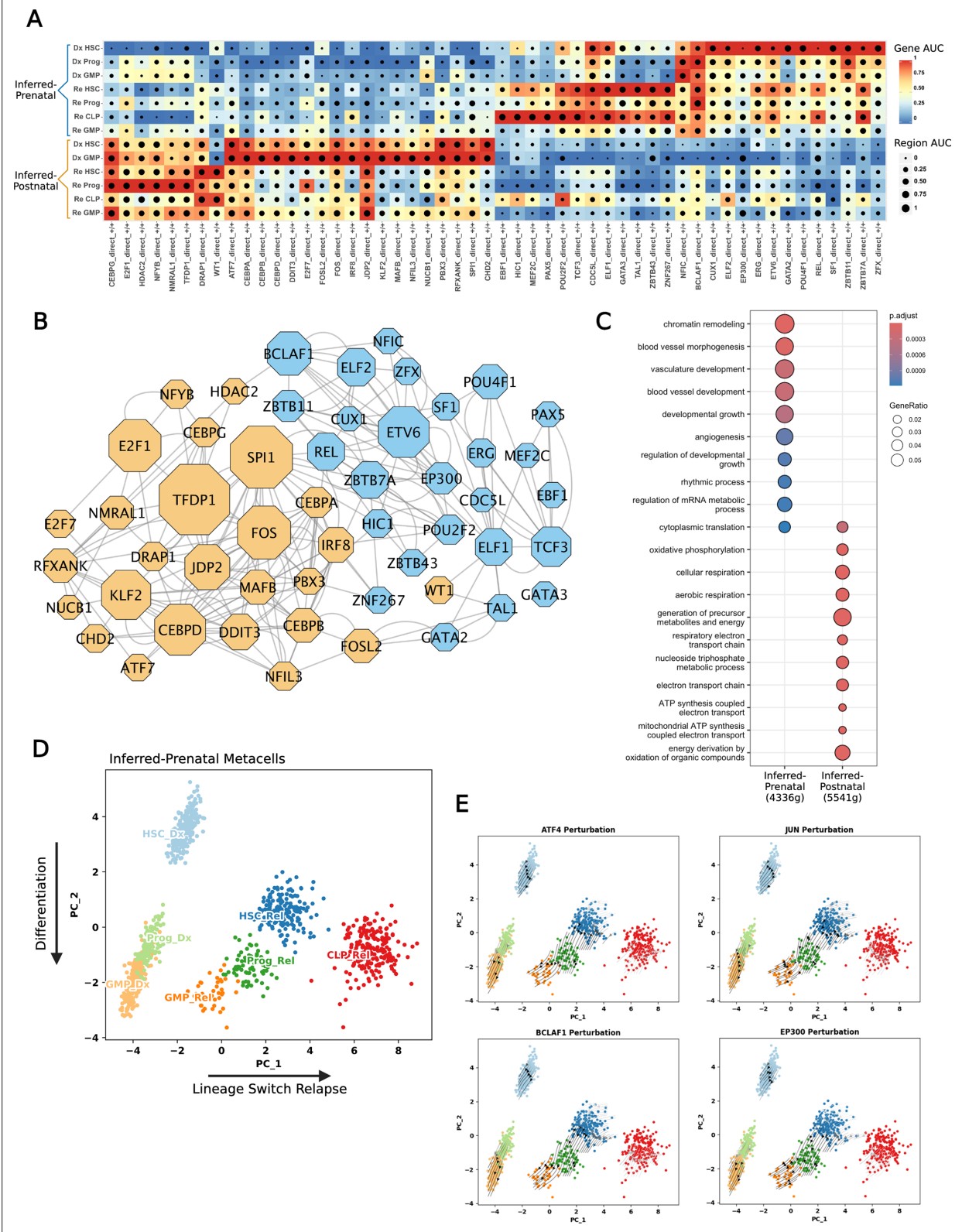

**Figure 5.** Combining multiomics data interrogates age-associated regulons. (**A**) SCENIC+ eRegulon dot plot of showing correlation between single-cell RNA sequencing (scRNA-seq) target gene activity (indicated by the color scale) and scATAC-seq target region accessibility (depicted by spot size). Regulon specificity score (RSS) identified the key activating eRegulons (+/+) between inferred-prenatal and inferred-postnatal origin disease and allows comparison of diagnosis (Dx) and relapse (Rel) time points. (**B**) Network showing the inferred-prenatal (blue) and inferred-postnatal (orange) associated

*Figure 5 continued on next page*

*Figure 5 continued*

eRegulons. Node size represents the number of target genes in each regulon. Edges represent interactions between nodes. (**C**) Over-representation analysis of age-associated eRegulon target genes using Gene Ontology (GO) Biological Processes curated gene sets. Adjusted p-value threshold 0.05. (**D**) Principal components analysis (PCA) of the gene based eRegulon enrichment scores for the inferred-prenatal origin disease at diagnosis and relapse. PC1 axis explains variance occurring between diagnosis and relapse, where this patient underwent a lineage switch. PC2 captures variance related to hematopoietic differentiation. (**E**) SCENIC+ perturbation simulation shows the predicted effect of knockout of selected transcription factors (TFs) on the previously computed PCA embedding. Arrows indicate the predicted shift in cell states relative to the initial PCA embedding.

The online version of this article includes the following source data and figure supplement(s) for figure 5:

**Source data 1.** SCENIC+ eRegulons for *Lambo et al., 2023* t(8;21) acute myeloid leukemia (AML) samples.

**Figure supplement 1.** Multiomics data of t(8;21) acute myeloid leukemia (AML) Rrefines gene regulatory network (GRN).

**Figure supplement 2.** SCENIC+ analysis identifies age-associated candidate perturbations in t(8;21) acute myeloid leukemia (AML).

## Discussion

Here, we have generated a new data resource, AML scAtlas, to investigate AML biology across a broad range of subtypes at single-cell resolution. By including 222 samples comprising 748,679 cells of patients with a wide range of clinical characteristics, AML scAtlas overcomes the limitations of many standalone single-cell studies enabling AML subtype-focused analysis with enough data for robust statistical comparisons. This dataset is publicly available (https://cellxgene.cziscience.com/collections/071b706a-7ea7-47a4-bddf-6457725839fc) providing the AML research community with a resource to address diverse biological questions and generate new hypotheses.

To further address a clinically relevant question using this data source, we compared differences between pediatric and adult-onset disease based on the potential biological effect of the in-utero origin of pediatric leukemia. Data of our AML scAtlas was used to explore the GRNs in adult and pediatric t(8;21) AML and revealed a strong age-associated GRN signature. This suggests that while pediatric and adult t(8;21) AMLs are propagated by the same driver translocation, they exhibit clear biological differences correlated with age. This may be due to differences in the cell-of-origin, with mouse models showing that t(8;21) AML can arise from a HSC or a more lineage-restricted GMP (*Cabezas-Wallscheid et al., 2013*). As pediatric t(8;21) can arise in-utero, as evidenced by previous studies (*Wiemels et al., 2002*), and adult t(8;21) is acquired postnatally (*Welch et al., 2012*; *Jaiswal et al., 2014*), we propose that the observed age-related differences in AML with t(8;21) reflect these differences in the developmental origins of the disease. We identified two distinct groups of regulons corresponding to either inferred-prenatal origin and inferred-postnatal origin disease. These regulons constitute the GRN underlying the cellular state, which can be informative when identifying molecular vulnerabilities to target leukemia.

Our cohort is the largest scRNA-seq dataset to explore t(8;21) AML biology to date, however, the number of patients included remains low (n=22), and many of the studies containing the adult samples used CD34 selection in their experimental protocol creating a bias towards HSPCs in these samples. To overcome some of these limitations, we used bulk RNA-seq samples from the TARGET (*Bolouri et al., 2018*) and BeatAML (*Burd et al., 2020*) studies with t(8;21) AML (n=83, *Source data 2*) to validate our regulon signatures. This identifies two clusters of samples which closely match these signatures, showing that the regulon patterns identified from our AML scAtlas are recapitulated with bulk RNA-seq data enabling exploration of larger patient cohorts. Comparisons between inferred-prenatal and inferred-postnatal origin transcriptomes prioritized TFs which were differentially expressed and highlighted differences in underlying biology and drug response. We identified five signature TFs (KDM5A, REST, BCLAF1, YY1, RAD21) for inferred-prenatal origin disease, several of which have roles in embryonic stem cells (*Singh et al., 2008*; *Dahl et al., 2016*), and critical functions in the maintenance of HSCs (*Lu et al., 2018*; *Fisher et al., 2017*; *Kumar et al., 2023*). In contrast, TFs identified in inferred-postnatal origin samples, such as interferon regulatory factors (IRFs), HDAC2, and SPI1, reflect inflammatory and immune processes, many of which have been implicated in leukemia (*Ning et al., 2011*; *Fischer et al., 2019*; *Fang et al., 2018*). We also found that inferred-prenatal origin samples had a higher proportion of HSC/Prog cell types compared to inferred-postnatal origin samples, a more primitive state than postnatal onset t(8;21) AML cases, supporting the hypothesis that age-associated differences in the cell-of-origin influence disease biology.

Given these biological differences, we used bulk RNA-seq to predict chemosensitivity using published drug response signatures (*Unnikrishnan et al., 2017*; *Williams et al., 2020*; *Xu et al., 2019*; *Zhang et al., 2020*). Inferred-prenatal samples were enriched for genes indicative of cytarabine sensitivity and depleted of genes suggestive of daunorubicin and venetoclax resistance. These findings suggest that the developmental origins of the disease may influence drug responses, with potential implications in the design of novel therapeutic strategies and providing further biological evidence that pediatric AML might benefit from different clinical management compared with adult-onset AML. Importantly, venetoclax is currently in the AML23 trial (NCT05955261); our results support further evaluation of venetoclax treatment in pediatric t(8;21) AML.

Using an additional single-cell multiomic dataset, using SCENIC+, we reconstructed the eGRN in samples matching our inferred-prenatal and inferred-postnatal regulon signatures. Upon comparing eRegulons for each patient at diagnosis and relapse, we identified clusters of highly correlated eRegulons defined by different biological processes. Inferred-prenatal origin samples are characterized by developmental and transcriptional dysregulation, whereas inferred-postnatal origin samples are largely driven by fundamental cellular processes linked to inflammation. We used SCENIC+ to model the predicted impact of TF perturbations on our prenatal origin sample at diagnosis and relapse identified several key components of the AP-1 complex, which are critical in t(8;21) AML biology and are also associated with dynamic age-related transcriptional changes (*Ptasinska et al., 2019*; *Martinez-Soria et al., 2018*; *Patrick et al., 2024*).

Through our analysis, we identified EP300 as a candidate target, which has been shown to be critical for t(8;21) AML biology (*Wang et al., 2011*) with demonstrable effects in KASUMI-1 and SKNO1 cell lines. EP300 has been identified as a promising therapeutic target in AML with several molecules in development (*Nicosia et al., 2023*) our data indicate potential specific therapeutic benefit in prenatal origin t(8;21) AML. One of the most impactful perturbation predictions for the HSC compartment at diagnosis and relapse was BCLAF1. This is consistent with previous evidence of its importance in HSCs (*Crowley et al., 2022a*; *Crowley et al., 2022b*) and AML (*Dell'Aversana et al., 2017*), but has not been studied specifically in the context of pediatric AML or t(8;21) AML previously. The DepMap data shows that KASUMI-1 is the most sensitive myeloid cell line to BCLAF1 perturbation, and our GRN analyses suggest it is particularly active in pediatric t(8;21) AML of inferred in-utero origin, thus this may represent an additional prognostic indicator.

Further investigations are required to characterize the roles of both EP300 and BCLAF1 in prenatal origin t(8;21) AML before any clinical realization. EP300 has already been investigated as a target in AML, so future work should focus on the pediatric AML setting with in vitro and in vivo studies using EP300/CBP inhibitors such as inobrodib (*Nicosia et al., 2023*). In contrast, BCLAF1 is relatively unexplored, and additional work is required to elucidate its molecular function and assess its potential as a therapeutic target. BCLAF1 may ultimately prove most valuable as a biomarker of in-utero t(8;21) AML, enabling distinction between late-onset in-utero and postnatal disease. This would require molecular validation in a large cohort of pediatric patients with Guthrie spots to confirm whether they had acquired t(8;21) in-utero.

## Conclusions

Overall, our study demonstrates that large-scale single-cell data integration is a powerful approach to dissect specific patient groups in detail and enables robust comparative analyses. We present the AML scAtlas as a publicly available resource for the research community to address diverse biological questions. By applying AML scAtlas to t(8;21) AML, we identified age-associated gene regulatory networks that likely reflect differences in the developmental origins, biology, and outcome of the disease. These findings also highlight novel candidate therapeutic targets which may be more relevant in pediatric t(8;21) AML compared to adult-onset disease, offering opportunities for more tailored treatment strategies.

## Methods

For the complete analysis code, including the conda environments used for analysis, see GitHub Repo (https://github.com/jesswhitts/AML-scAtlas, copy archived at *Whittle, 2026*).

## Data collection

A literature search was performed for published AML scRNA-seq datasets (*Velten et al., 2021*; *van Galen et al., 2019*; *Beneyto-Calabuig et al., 2023*; *Stetson et al., 2021*; *Zheng et al., 2017*; *Petti et al., 2019*; *Jiang et al., 2020*; *Johnston et al., 2020*; *Pei et al., 2020*; *Li et al., 2023*; *Lasry et al., 2023*; *Fiskus et al., 2023*; *Naldini et al., 2023*; *Mumme et al., 2023*; *Zhang et al., 2023*; *Li et al., 2022*; *Oetjen et al., 2018*; *Setty et al., 2019*; *Caron et al., 2020*). Suitable studies were selected based on the data quality (over 1000 counts and 500 genes detected per cell for most of the data). Diagnostic, primary AML samples were selected from each AML study. Where healthy donor samples were present, these were also included, along with an additional 4 studies with healthy bone marrow samples.

## Initial data processing

Each scRNA-seq dataset underwent initial quality control individually using Scanpy (v1.9.3) (*Wolf et al., 2018*) as some studies provided raw data and others provided pre-filtered data. Where raw data was provided, doublets were removed using Scrublet (v0.2.3) (*Wolock et al., 2019*) and cells were filtered using the median absolute deviation as described in this single-cell best practices hand-book (*Heumos et al., 2023*; *Heumos and Schaar, 2023*).

Once filtered, datasets were combined, and quality control was performed using Scanpy (v1.9.3) (*Wolf et al., 2018*). The full dataset had quality thresholds applied (percentage mitochondrial counts <10, read counts >1000, gene counts >500), removing any samples which had fewer than 50 cells remaining after filtering. Genes present in <50 cells were removed. MALAT1 was removed as this was highly abundant in many cells and considered artefactual.

## Batch correction

The presence of batch effects was determined through dimensionality reduction and clustering using Scanpy (v1.9.3) (*Wolf et al., 2018*) and using the kBET algorithm (v0.99.6) (*Büttner et al., 2019*). This was repeated on individual studies, to assess whether there were sample-wise batch effects. Batch correction benchmarking was implemented using Harmony (Scanpy v1.9.3 implementation) (*Korsunsky et al., 2019*), scVI (v1.0.3) (*Lopez et al., 2018*), and scANVI (v1.0.3) (*Xu et al., 2021*) and quantified using scIB (1.1.4) (*Luecken et al., 2022*). Different numbers of highly variable genes were used to select the optimal number for integration. Batch correction was performed using scVI (v1.0.3) (*Lopez et al., 2018*) with the top 2000 highly variable genes, using sample as the model covariate.

## AML scAtlas cell type annotation

The scVI corrected embedding was used to run UMAP and Leiden clustering using Scanpy functions (v1.9.3) (*Wolf et al., 2018*). Cell type annotation was performed using CellTypist (v1.6.0) (*Domínguez Conde et al., 2022*) using the 'Immune_All_Low.pkl' model, SingleR (v2.0.0) (*Aran et al., 2019*) using the Novershtern hematopoietic reference (*Novershtern et al., 2011*), and scType (v1.0) (*Ianevski et al., 2022*) with the tissue defined as 'Immune system.' Full automated tool outputs are detailed in *Figure 1—source data 2*; overall, we found that the results given varied significantly between different tools. We postulate that this is, in part, due to differences in the reference profiles used. Thus, we opted to use the best consensus of these different tools for our cluster identity assignments.

## AML scAtlas LSC annotation

HSPC clusters were selected from AML scAtlas, and the scVI corrected embedding was used to recompute UMAP using Scanpy functions (v1.9.3) (*Wolf et al., 2018*). As our previous cell type annotations used generic reference profiles and were not AML-specific, we generated a custom cell type annotation reference to identify LSCs. We created a custom SingleR (v2.0.0) (*Aran et al., 2019*) reference using the *Zeng et al., 2022* revised annotations of the *van Galen et al., 2019* dataset (*Figure 1—source data 2*). This was also correlated with the LSC6 (*Elsayed et al., 2020*) and LSC17 (*Ng et al., 2016*) scores for each cell. To compare LSC abundance between ELN risk groups, chi2_contingency was implemented from SciPy (v1.12.0).

## AML with t(8;21) analysis

Samples with the t(8;21) translocation were selected from the full AML scAtlas. The UMAP was re-computed, and genes were filtered to remove those detected in fewer than 50 cells for the revised dataset, leaving 24,866 genes remaining. Gene regulatory network analysis was performed using pySCENIC (*Aibar et al., 2017*; *Van de Sande et al., 2020*) (v0.12.1) as per the recommended workflow. To facilitate comparisons between age groups, cell types were focused on HSPCs, as many adult samples were originally enriched for CD34. The RSS was calculated for the adult and pediatric samples to select the top 20 differential regulons per age group. Using SciPy hierarchical clustering (v1.12.0), regulons were filtered to identify regulon signature groups used for downstream analysis.

## Bulk RNA-seq analysis

Bulk RNA-Seq data was downloaded for the TARGET (*Bolouri et al., 2018*) and BeatAML (*Burd et al., 2020*) cohorts and samples with t(8;21) were selected. Only bone marrow samples taken at diagnosis were used for downstream analyses (*Source data 2*). Using the previously defined signature regulons, the AUCell algorithm (*Aibar et al., 2017*) (v0.12.1) was implemented to measure regulon activity. Hierarchical clustering was performed using SciPy v1.12.0 to identify samples most enriched for each age-related signature. Differential gene expression analysis was implemented using edgeR (*Robinson et al., 2010*) (v3.42.4) and DESeq2 (*Love et al., 2014*) (v1.40.2), using a log2 fold change threshold of 0.5 and an adjusted p-value cutoff of 0.01. Candidate differential genes, ranked on log2 fold change, underwent GSEA with GSEApy (v0.10.8) using a significance threshold of 0.05. Cell type deconvolution was performed using AutoGeneS (*Aliee and Theis, 2021*) (v1.0.4) using the recommended workflow. Significance when comparing groups was ascertained using a Student's t-test on the predicted cell type proportion values for each sample.

## Single-cell multi-omics analysis

The *Lambo et al., 2023* scRNA-seq and scATAC-seq data from pediatric AML bone marrow samples was downloaded, and the t(8;21) samples selected (*Source data 2*). Using our previously defined signature regulons, the AUCell algorithm (*Aibar et al., 2017*) (v0.12.1) was implemented to measure regulon activity. This identified the samples most enriched for each age-related signature as AML16 and AML12.

The SCENIC+ pipeline (*Bravo González-Blas et al., 2023*) (v1.0a1) was implemented as per the recommended Snakemake workflow for creating pseudo-multiome data. Regulons were filtered by correlation between modalities, using a threshold of 0.2 for non-multiome data. The most robust regulons were prioritized based on the SCENIC+ recommendations (direct +/+) (*Bravo González-Blas et al., 2023*). To facilitate comparisons, the eRegulon RSS was calculated for each patient and the top 30 eRegulons selected. The correlation between the gene sets underpinning eRegulons was calculated and sample-associated clusters were selected for over-representation analysis with clusterProfiler (*Wu et al., 2021*) (v4.8.3).

To predict the impact of specific TF perturbations on key cell types, SCENIC+ (*Bravo González-Blas et al., 2023*) perturbation modelling was implemented using the recommended parameters. TFs were then prioritized on their predicted impact on HSC differentiation and visualized using the PCA embedding. Candidate targets EP300 and BCLAF1 were queried in the DepMap (*Tsherniak et al., 2017*) databases to infer their potential importance.

## Acknowledgements

JW was funded by MRC DTP award (MR/W007428/1), SM by Blood Cancer UK (15038), and CCLG (2016 09), while GL by Cancer Research UK (C5759/A20971 & C5759/A27412) and MI by MRC (MR/X014088/1). We would like to thank all authors of the public data used in this study for their contributions to scientific community. We also acknowledge useful discussions around this work with Magnus Rattray.

## Additional information

### Funding

| Funder | Grant reference number | Author |
| --- | --- | --- |
| Medical Research Council | MR/W007428/1 | Jessica Whittle |
| Blood Cancer UK | 15038 | Stefan Meyer |
| Children's Cancer and Leukaemia Group | 2016 09 | Stefan Meyer |
| Cancer Research UK | C5759/A20971 | Georges Lacaud |
| Cancer Research UK | C5759/A27412 | Georges Lacaud |
| Blood Cancer UK | 23002 | Georges Lacaud |
| Medical Research Council | MR/X014088/1 | Mudassar Iqbal |

The funders had no role in study design, data collection and interpretation, or the decision to submit the work for publication.

### Author contributions

Jessica Whittle, Conceptualization, Data curation, Formal analysis, Writing – original draft, Writing – review and editing; Stefan Meyer, Conceptualization, Formal analysis, Writing – original draft, Writing – review and editing; Georges Lacaud, Syed Murtuza-Baker, Mudassar Iqbal, Conceptualization, Formal analysis, Supervision, Funding acquisition, Writing – original draft, Project administration, Writing – review and editing

### Author ORCIDs

Jessica Whittle ⓘ https://orcid.org/0000-0002-1139-1671
Georges Lacaud ⓘ https://orcid.org/0000-0002-5630-2417
Mudassar Iqbal ⓘ https://orcid.org/0000-0002-5006-4331

Reviewer #1 (Public review): https://doi.org/10.7554/eLife.104978.3.sa1
Author response https://doi.org/10.7554/eLife.104978.3.sa2

---

## Additional files

### Supplementary files

MDAR checklist

Source data 1. Datasets included in acute myeloid leukemia (AML) single-cell Atlas (scAtlas).

Source data 2. External validation datasets used in this study.

### Data availability

The AML scAtlas is hosted online for public use (https://cellxgene.cziscience.com/collections/071b706a-7ea7-47a4-bddf-6457725839fc). The processed AnnData object is also available to download from figshare (https://doi.org/10.48420/27269946.v2). Details of all data used in this study can be found in *Source data 1*, along with associated links to the original data. All code used to perform the analyses presented here can be accessed in the GitHub repository: (https://github.com/jesswhitts/AML-scAtlas, copy archived at *Whittle, 2026*). The samples used for validation analyses are publicly available and are detailed in *Source data 2*. The SCENIC+ eGRN files are provided as *Figure 5—source data 1*.

The following dataset was generated:

| Author(s) | Year | Dataset title | Dataset URL | Database and Identifier |
|---|---|---|---|---|
| Whittle J, Meyer S, Lacaud G, Baker SM, Iqbal M | 2025 | A Single-Cell Transcriptomic Atlas of Acute Myeloid Leukemia | https://doi.org/10.48420/27269946 | figshare, 10.48420/27269946 |

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
