## [Editor Report · eLife Assessment]

This manuscript provides a single-cell transcriptomic atlas for AML (222 samples comprising 748,679 cells) integrating data from multiple studies. They use this dataset to investigate t(8;21) AML, and they reconstruct the Gene Regulatory Network and enhancer Gene Regulatory Network, which allowed identification of interesting targets. This aggregation is **important** and can help infer differences in genetic regulatory modules based on the age of disease onset. Their **compelling** effort may help explain age-related variations in prognosis and disease development in subtype-specific manner.

---

## [Referee Report · Reviewer #1 (Public review)]

Summary:

In this manuscript, the authors performed an integration of 48 scRNA-seq public datasets and created a single-cell transcriptomic atlas for AML (222 samples comprising 748,679 cells). This is important since most AML scRNA-seq studies suffer from small sample size coupled with high heterogeneity. They used this atlas to further dissect AML with t(8;21) (AML-ETO/RUNX1-RUNX1T1), which is one of the most frequent AML subtypes in young people. In particular, they were able to predict Gene Regulatory Networks in this AML subtype using pySCENIC, which identified the paediatric regulon defined by a distinct group of hematopoietic transcription factors (TFs) and the adult regulon for t(8;21). They further validated this in bulk RNA-seq with AUCell algorithm and inferred prenatal signature to 5 key TFs (KDM5A, REST, BCLAF1, YY1, and RAD21), and the postnatal signature to 9 TFs (ENO1, TFDP1, MYBL2, KLF1, TAGLN2, KLF2, IRF7, SPI1, and YXB1). They also used SCENIC+ to identify enhancer-driven regulons (eRegulons), forming an eGRN, and found that prenatal origin shows a specific HSC eRegulon profile, while a postnatal shows a GMP profile. They also did an in silico perturbation and found AP-1 complex (JUN, ATF4, FOSL2), P300 and BCLAF1 as important TFs to induce differentiation. Overall, I found this study very important in creating a comprehensive resource for AML research.

Strengths:

• The generation of an AML atlas integrating multiple datasets with almost 750K cells will further support the community working on AML

• Characterisation of t(8;21) AML proposes new interesting leads.

• The t(8;21) TFs/regulons identified from any of the single dataset are not complete and now the authors showed that the increase in the number of cells that allowed identification of novel ones.

Comments on revisions:

In the revised version of the manuscript, the authors addressed all my comments.

---

## [Author Response]

The following is the authors’ response to the original reviews.

**Reviewer #1 (Public review):**
Summary:In this manuscript, the authors performed an integration of 48 scRNA-seq public datasets and created a single-cell transcriptomic atlas for AML (222 samples comprising 748,679 cells). This is important since most AML scRNA-seq studies suffer from small sample size coupled with high heterogeneity. They used this atlas to further dissect AML with t(8;21) (AML-ETO/RUNX1-RUNX1T1), which is one of the most frequent AML subtypes in young people. In particular, they were able to predict Gene Regulatory Networks in this AML subtype using pySCENIC, which identified the paediatric regulon defined by a distinct group of hematopoietic transcription factors (TFs) and the adult regulon for t(8;21). They further validated this in bulk RNA-seq with AUCell algorithm and inferred prenatal signature to 5 key TFs (KDM5A, REST, BCLAF1, YY1, and RAD21), and the postnatal signature to 9 TFs (ENO1, TFDP1, MYBL2, KLF1, TAGLN2, KLF2, IRF7, SPI1, and YXB1). They also used SCENIC+ to identify enhancer-driven regulons (eRegulons), forming an eGRN, and found that prenatal origin shows a specific HSC eRegulon profile, while a postnatal origin shows a GMP profile. They also did an in silico perturbation and found AP-1 complex (JUN, ATF4, FOSL2), P300, and BCLAF1 as important TFs to induce differentiation. Overall, I found this study very important in creating a comprehensive resource for AML research.Strengths:(1) The generation of an AML atlas integrating multiple datasets with almost 750K cells will further support the community working on AML.(2) Characterisation of t(8;21) AML proposes new interesting leads.

We thank the reviewer for a succinct summary of our work and highlighting its strengths.

Weaknesses:Were these t(8;21) TFs/regulons identified from any of the single datasets? For example, if the authors apply pySCENIC to any dataset, would they find the same TFs, or is it the increase in the number of cells that allows identification of these?

We implemented pySCENIC on individual datasets and compared the TFs (defining the regulons) identified to those from the combined AML scAtlas analysis. There were some common TFs identified, but these vary between individual studies. The union of all TFs identified makes a very large set - comprising around a third of all known TFs. AML scAtlas provides a more refined repertoire of TFs, perhaps as the underlying network inference approach is more robust with a higher number of cells. The findings of these investigations are included in Supplementary Figure 4DE, we hope this is useful for other users of pySCENIC.

**Reviewer #2 (Public review):**
Summary:The authors assemble 222 publicly available bone marrow single-cell RNA sequencing samples from healthy donors and primary AML, including pediatric, adolescent, and adult patients at diagnosis. Focusing on one specific subtype, t(8;21), which, despite affecting all age classes, is associated with better prognosis and drug response for younger patients, the authors investigate if this difference is reflected also in the transcriptomic signal. Specifically, they hypothesize that the pediatric and part of the young population acquires leukemic mutations in utero, which leads to a different leukemogenic transformation and ultimately to differently regulated leukemic stem cells with respect to the adult counterpart. The analysis in this work heavily relies on regulatory network inference and clustering (via SCENIC tools), which identifies regulatory modules believed to distinguish the pre-, respectively, post-natal leukemic transformation. Bulk RNA-seq and scATAC-seq datasets displaying the same signatures are subsequently used for extending the pool of putative signature-specific TFs and enhancer elements. Through gene set enrichment, ontology, and perturbation simulation, the authors aim to interpret the regulatory signatures and translate them into potential onset-specific therapeutic targets. The putative pre-natal signature is associated with increased chemosensitivity, RNA splicing, histone modification, stemness marker SMARCA2, and potentially maintained by EP300 and BCLAF1.Strengths:The main strength of this work is the compilation of a pediatric AML atlas using the efficient Cellxgene interface. Also, the idea of identifying markers for different disease onsets, interpreting them from a developmental angle, and connecting this to the different therapy and relapse observations, is interesting. The results obtained, the set of putative up-regulated TFs, are biologically coherent with the mechanisms and the conclusions drawn. I also appreciate that the analysis code was made available and is well documented.

We thank the reviewer for evaluating our work, and highlighting its key features, including creation of AML atlas, downstream analysis and interpretation for t(8;21) subtype.

Weaknesses:There were fundamental flaws in how methods and samples were applied, a general lack of critical examination of both the results and the appropriateness of the methods for the data at hand, and in how results were presented. In particular:(1) Cell type annotation:(a) The 2-phase cell type annotation process employed for the scRNA-seq sample collection raised concerns. Initially annotated cells are re-labeled after a second round with the same cell types from the initial label pool (Figure 1E). The automatic annotation tools were used without specifying the database and tissue atlases used as a reference, and no information was shown regarding the consensus across these tools.

Cell type annotations are heavily influenced by the reference profiles used and vary significantly between tools. To address this, we used multiple cell type annotation tools which predominantly encompassed healthy peripheral blood cell types and/or healthy bone marrow populations. This determined the primary cluster cell types assigned.

Existing tools and resources are not leukemia specific, thus, to identify AMLassociated HSPC subpopulations we created a custom SingleR reference, using a CD34 enriched AML single-cell dataset. This was not suitable for the annotation of the full AML scAtlas, as it is derived from CD34 sorted cell types so is biased towards these populations.

We have made this much clearer in the revised manuscript, by splitting Figure 1 into two separate figures (now Figure 1 and Figure 2) reflecting both different analyses performed. The methods have also been updated with more detail on the cell type annotations, and we have included the automated annotation outputs as a supplementary table, as this may be useful for others in the single-cell community.

(b) Expression of the CD34 marker is only reported as a selection method for HSPCs, which is not in line with common practice. The use of only is admitted as a surface marker, while robust annotation of HSPCs should be done on the basis of expression of gene sets.

Most of the cells used in the HSPC analysis were in fact annotated as HSPCs with some exceptions. In line with this feedback, we have re-worked this analysis and simply taken HSPC annotated clusters forward for the subsequent analysis, yielding the same findings.

(c) During several analyses, the cell types used were either not well defined or contradictory, such as in Figure 2D, where it is not clear if pySCENIC and AUC scores were computed on HSPCs alone or merged with CMPs. In other cases, different cell type populations are compared and used interchangeably: comparing the HSPCderived regulons with bulk (probably not enriched for CD34+ cells) RNA samples could be an issue if there are no valid assumptions on the cell composition of the bulk sample.

We apologize for the lack of clarity regarding which cell types were used, the text has been updated to clarify that in the pySCENIC analysis all myeloid progenitor cells were included.

The bulk RNA-seq samples were used only to test the enrichment of our AML scAtlas derived regulons in an unbiased and large-scale way. While CD34 enriched samples could be preferable, this was not available to us.

We agree that more effort could be made to ensure the single-cell/myeloid progenitor derived regulons are comparable to the bulk-RNA sequencing data. In the original bulk RNA-seq validation analysis, we used all bulk-RNA sequencing timepoints (diagnostic, on-treatment, relapse) and included both bone marrow and peripheral blood. Upon reflection, and to better harmonize the bulk RNA-seq selection strategy with that of AML scAtlas, we revised our approach to include only diagnostic bone marrow samples. We expect that, since the leukemia blast count for pediatric AML is typically high at diagnosis, these samples will predominantly contain leukemic blasts.

(2) Method selection:(a) The authors should explain why they use pySCENIC and not any other approach.They should briefly explain how pySCENIC works and what they get out in the main text. In addition they should explain the AUCell algorithm and motivate its usage.

pySCENIC is state-of-the-art method for network inference from scRNA data and is widely used within the single-cell community (over 5000 citations for both versions of the SCENIC pipeline). The pipeline has been benchmarked as one of the top performers for GRN analysis (Nguyen et al, 2021. Briefings in Bioinformatics). AUCELL is a module within the pySCENIC pipeline to summarize the activity of a set of genes (a regulon) into a single number which helps compare and visualize different regulons. We have modified the manuscript (Results section 2 paragraph 2) to better explain this method and provided some rationale and accompanying citations to justify its use for this analysis. We thank the reviewer for highlighting this and hope our updates add some clarity.

(b) The obtained GRN signatures were not critically challenged on an external dataset. Therefore, the evidence that supports these signatures to be reliable and significant to the investigated setting is weak.

These signatures were inferred using the most suitable AML single-cell RNA datasets currently available. To validate our findings, we used two independent datasets (the TARGET AML bulk RNA sequencing cohort, and the Lambo et al. scRNA-seq dataset). To clarify this workflow in the manuscript, we have added a panel to Figure 3 outlining the analytical process. To our knowledge, there are no other better-suited datasets for validation. Experimental validations on patient samples, while valuable, are beyond the scope of this study.

(3) There are some issues with the analysis & visualization of the data.

Based on this feedback, we have improved several aspects of the analysis, changed some visualizations, and improved figure resolution throughout the manuscript.

(4) Discussion:(a) What exactly is the 'regulon signature' that the authors infer? How can it be useful for insights into disease mechanisms?

The ’regulon signature’ here refers to a gene regulatory program (multiple gene modules, each defined by a transcription factor and its targets) which are specific to different age groups. Further investigation into this can be useful for understanding why patients of different ages confer a different clinical course. We have amended the text to explain this.

(b) The authors write 'Together this indicates that EP300 inhibition may be particularly effective in t(8;21) AML, and that BCLAF1 may present a new therapeutic target for t(8;21) AML, particularly in children with inferred pre-natal origin of the driver translocation.' I am missing a critical discussion of what is needed to further test the two targets. Put differently: Would the authors take the risk of a clinical study given the evidence from their analysis?

Indeed, many extensive studies would be required before these findings are clinically translatable. We have included a discussion paragraph (discussion paragraph 7) detailing what further work is required in terms of experimental validation and potential subsequent clinical study.

**Reviewer #1 (Recommendations for the authors):**
In addition to the point raised above, Cytoscape files for the GRNs and eGRNs inferred would be useful to have.

We have now provided Cytoscape/eGRN tables in supplementary materials.

**Reviewer #2 (Recommendations for the authors):**
(1) Figures 1F and 1G: You show the summed-up frequencies for all patients, right? It would be very interesting to see this per patient, or add error bars, since the shown frequencies might be driven by single patients with many cells.

While this type of plot could be informative, the large number of samples in the AML scAtlas rendered the output difficult to interpret. As a result, we decided not to include it in the manuscript.

(2) An issue of selection bias has to be raised when only the two samples expressing the expected signatures are selected from the external scRNA dataset. Similarly, in the DepMap analysis, the age and nature of the other cell lines sensitive to EP300 and BCLAF1 should be reported.

Since the purpose of this analysis was to build on previously defined signatures, we selected the two samples which we had preliminary hypotheses for. It would indeed be interesting to explore those not matching these signatures; however, samples numbers are very small, so without preliminary findings robust interpretation and validation would be difficult. An expanded validation would be more appropriate once more data becomes available in the future.

We agree that investigating the age and nature of other BCLAF1/EP300 sensitive cell lines is a very valuable direction. Our analysis suggests that our BCLAF1 findings may also be applicable to other in-utero origin cancers, and we have now summarized these observations in Supplementary Figure 7H.

(3) Is there statistical evidence for your claim that "This shows that higher-risk subtypes have a higher proportion of LSCs compared to favorable risk disease."? At least intermediate and adverse look similar to me. How does this look if you show single patients?

We are grateful to the reviewer for noticing this oversight and have now included an appropriate statistical test in the revised manuscript. As before, while showing single patients may be useful, the large number of patients makes such plot difficult to interpret. For this reason, we have chosen not to include them.

(4) Specify the statistical test you used to 'identify significantly differentially expressed TFs' (line 192).

The methods used for differential expression analysis are now clearly stated in the text as well as in the methods section. We hope this addition improves clarity for the reader.

(5) Figure 2B: You show the summed up frequencies for all patients, right? It would be intriguing to see this figure per patient, since the shown frequencies might be driven by single patients with many cells.

Yes, the plot includes all patients. Showing individual patients on a single plot is not easily interpretable.

(6) Y axis in 2D is not samples, but single cells? Please specify.

We thank the reviewer for bringing this to our attention and have now updated Figure 3D accordingly.

(7) Figure 3A: I don't get why the chosen clusters are designated as post- and prenatal, given the occurrence of samples in them.

This figure serves to validate the previously defined regulon signatures, so the cluster designations are based on this. We have amended the text to elaborate on this point, which will hopefully provide greater clarity.

(8) Figure 3E: What is shown on the y axis? Did you correct your p-values for multiple testing?

We apologize for this oversight and have now added a y axis label. P values were not corrected for multiple testing, as there are only few pairwise T tests performed.

(9) Robustness: You find some gene sets up- and down-regulated. How would that change if you used an eg bootstrapped number of samples, or a different analysis approach?

To address this, we implemented both edgeR and DESeq2 for DE testing. Our findings (Supplementary Figure 5B) show that 98% of edgeR genes are also detected by DESeq2. We opted to use the smaller edgeR gene list for our analysis, due to the significant overlap showing robust findings. We thank the reviewer for this helpful suggestion, which has strengthened our analysis

(10) Multiomics analysis:(a) Why only work on 'representative samples'? The idea of an integrated atlas is to identify robust patterns across patients, no? I'd love to see what regulons are robust, ie, shared between patients.

As discussed in point 2, there are very few samples available for the multiomics analysis. Therefore, we chose to focus on those samples which we had a working hypothesis for, as a validation for our other analyses.

(b) I don't agree that finding 'the key molecular processes, such as RNA splicing, histone modification, and TF binding' expressed 'further supports the stemness signature in presumed prenatal origin t(8;21) AML'.

Following the improvements made on the bulk RNA-Seq analysis in response to the previous reviewer comments, we ended up with a smaller gene set. Consequently, the ontology results have changed. The updated results are now more specific and indicate that developmental processes are upregulated in presumed prenatal origin t(8;21) AML.

(c) Please clarify if the multiome data is part of the atlas.

The multiome data is not a part of AML scAtlas, as it was published at a later date. We used this dataset solely for validation purposes and have updated the figures and text to clearly indicate that it is used as a validation dataset.

(d) Please describe the used data with respect to the number of patients, cells, age, etc.

We clarified this point in the text and have also included supplementary tables detailing all samples used in the atlas and validation datasets.

(e) The four figures in Figure 4E look identical to me. What is the take-home message here? Do all perturbations have the same impact on driving differentiation? Please elaborate.

The perturbation figure is intended to illustrate that other genes can behave similarly to members of the AP-1 complex (JUN and ATF4 here) following perturbation. Since the AP-1 complex is well known to be important in t(8;21) AML, we hypothesize that these other genes are also important. We apologize for the previous lack of interpretation here and have amended the text to clarify this point.

(11) Abstract: Please detail: how many of the 159 AML patients are t(8;21)?

We have now amended the abstract to include this.

(12) Figures: Increase font size where possible, eg age in 1B or risk group in 1G is super small and hard to read.

Extra attention has been given to improving the figure readability and resolution throughout the whole manuscript.

(13) Color codes in Figures 2B and 2C are all over the place and misleading: Sort 2C along age, indicate what is adult and adolescent, sort the x axis in 2B along age.

We have changed this figure accordingly.

(14) I suggest not coloring dendrograms, in my opinion this is highly irritating.

The dendrogram colors correspond to clusters which are referenced in the text, this coloring provides informative context and aids interpretation, making it a useful addition to the figure.

(15) The resolution in Figure 4B is bad, I can't read the labels.

This visualization has been revised, to make presentation of this data clearer.

(16) In addition to selecting bulk RNA samples matching the two regulon signatures, some effort should have been put into investigating the samples not aligned with those, or assessing how unique these GRN signatures are to the specific cell type and disease of interest, excluding the influence of cell type composition and random noise. The lateonset signatures should also be excluded from being present in an external pre-natal cohort in a more statistically rigorous manner.

Our use of the bulk RNA-Seq data is solely intended for the validation of predefined regulon signatures, for which we already have a working hypothesis. While we agree that further investigation of the samples that do not align with these signatures could yield interesting insights, we believe that such an analysis would extend beyond the scope of the current manuscript.

(17) The specific bulk RNA samples used should be specified, along with the tissue of origin. The same goes for the Lambo dataset.

We have clarified this point in the text and provided a supplementary table detailing all samples used for validation, alongside the sample list from AML scAtlas.

(18) In Supplementary Figure 5 B, the axes should be define.

We have updated this figure to include axis legends.

(19) Supplementary Figure 4A. There is a mistake in the sex assignment for sample AML14D. Since chrY-genes are expressed, this sample is likely male, while the Xist expression is mostly zero.

We thank the reviewer for pointing out this error, which has now been corrected.

(20) Wording suggestions:(a) Line 54: not compelling phrasing.(b) Line 83: "allows to decipher".(c) Line 88: repetition from line 85.(d) Line 90: the expression "clean GRN" is not clear.

These wording suggestions have all been incorporated in the revised manuscript.

(21) Supplementary Figure 3D is not interpretable, I suggest a different visualization.

We agree that the original figure was not the most informative and have replaced it with UMAPs displaying LSC6 and LSC17 scores.